# ItinBench: Benchmarking Planning Across Multiple Cognitive Dimensions with Large Language Models

## Abstract

Large language models (LLMs) with advanced cognitive capabilities are emerging as agents for various reasoning and planning tasks. Traditional evaluations often focus on specific reasoning or planning questions within controlled environments. Recent studies have explored travel planning as a medium to integrate various verbal reasoning tasks into real-world contexts. However, reasoning tasks extend beyond verbal reasoning alone, and a comprehensive evaluation of LLMs requires a testbed that incorporates tasks from multiple cognitive domains. To address this gap, we introduce ItinBench, a benchmark that features one task of spatial reasoning, i.e., route optimization, into trip itinerary planning while keeping the traditional verbal reasoning tasks. ItinBench evaluates various LLMs across diverse tasks simultaneously, including Llama 3.1 8B, Mistral Large, Gemini 1.5 Pro, and GPT family. Our findings reveal that LLMs struggle to maintain high and consistent performance when concurrently handling multiple cognitive dimensions. By incorporating tasks from distinct human-level cognitive domains, ItinBench provides new insights into building more comprehensive reasoning testbeds that better reflect real-world challenges. The code and dataset are attached.

## 1 Introduction

Building on LLMs' foundational Natural Language Processing (NLP) capabilities such as translation, text generation, and conversational interaction (Donthi et al., 2025; Hong et al., 2025; Plaat et al., 2024), LLMs demonstrate remarkable proficiency in various reasoning tasks (Ferrag et al., 2025; Lai et al., 2024; Du et al., 2024) and are evaluated in the corresponding benchmarks (Guo et al., 2025; Wang et al., 2024b; Li et al., 2024; Wang et al., 2024c). This progress lays the groundwork for their applications in various planning scenarios (Zhao et al., 2024; Valmeekam et al., 2024; Ruan et al., 2023). One drawback of these benchmarks is that the experiments are often limited in predefined settings, deterministic ground truths, and tasks confined to specific reasoning domains. In response to these concerns, new benchmarks, datasets, and related models have emerged (Xie et al., 2024; Hao et al., 2024; Tang et al., 2024; Kambhampati et al., 2024). They aim to create realistic sandboxes and develop language agents capable of performing complex reasoning and planning tasks. However, most evaluations still emphasize linguistic, logical, and mathematical reasoning—i.e. verbal reasoning (Polk, 1992).

Human-level cognition extends beyond verbal reasoning to include spatial reasoning—a core aspect of human intelligence (Whiteley et al., 2015). Spatial reasoning plays a vital role in various planning tasks—ranging from explicit activities such as navigating unfamiliar environments and organizing travel routes (Levinson, 2003), to more implicit ones like analyzing sports strategies or efficiently packing a backpack. While spatial reasoning is broad and multifaceted, we do not attempt to exhaustively evaluate all of its forms; instead, we focus on a representative spatial subtask that integrates seamlessly into the travel-planning setting—route optimization—allowing joint evaluation alongside traditional verbal reasoning. The non-symbolic nature of spatial reasoning makes it significantly less overlap with verbal reasoning abilities. It requires a more abstract "imagination" in the "brain" capability (Wu et al., 2024b). Given the pervasive role of spatial reasoning in human cognition, it raises several challenging and meaningful real-world questions. When spatial reasoning and planning are integral to complex verbal reasoning tasks, can LLMs perform well on spatial sub-tasks as

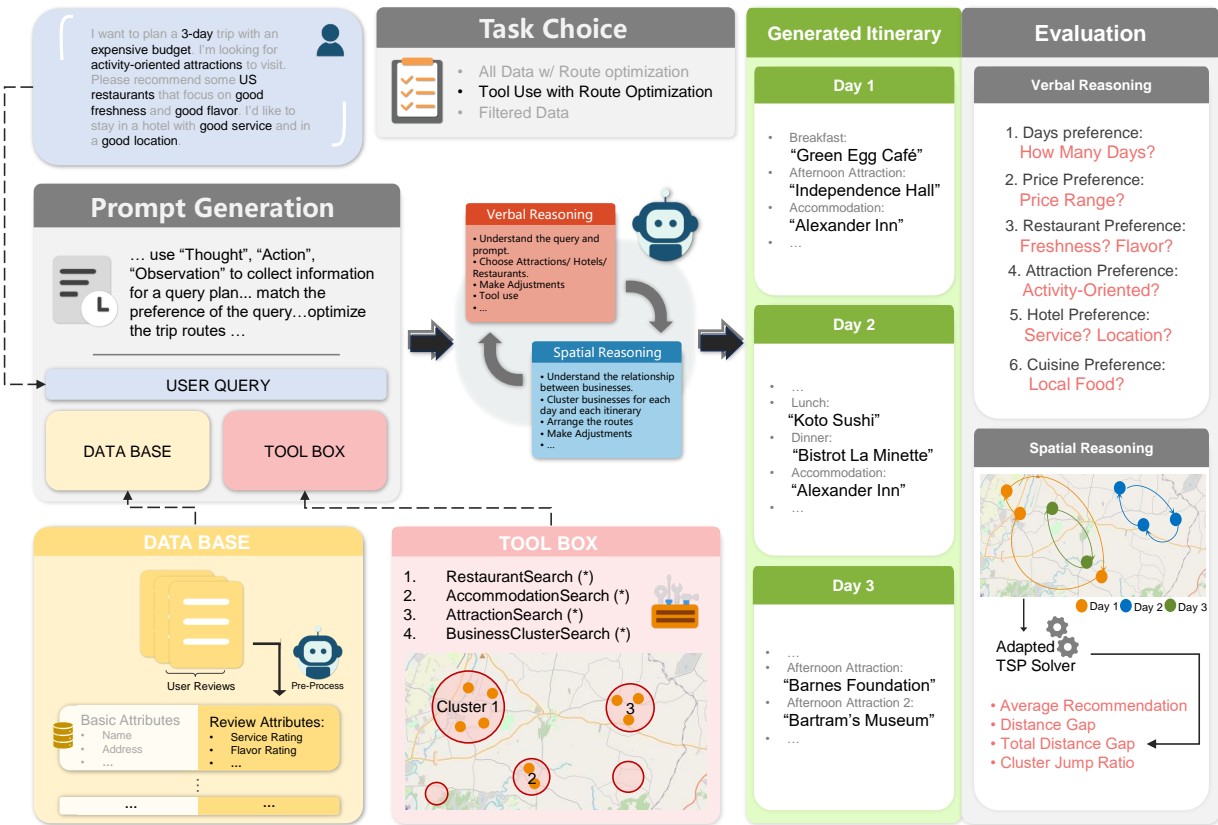

Figure 1: An overview of ItinBench. One of the four tasks, "Tool Use with Route Optimization," is chosen in this figure. The database with additional extracted information from user reviews, the human query, and a list of tools are integrated into a task-specific prompt. LLMs need to utilize their verbal and spatial reasoning ability to plan a trip itinerary based on the task constructed. The verbal and spatial reasoning aspects are evaluated to assess LLMs' ability to simultaneously address tasks from multiple cognitive dimensions.

they do on verbal ones? Furthermore, is there a performance trade-off when LLMs are required to handle both verbal and spatial reasoning simultaneously?

Thus, we propose ItinBench. A benchmark that expands the evaluated cognitive dimensions from sole verbal reasoning to spatial reasoning. As described in Figure 1, this facilitates the downstream task—trip itinerary planning. The pipeline contains the prompts for different tasks which integrating user query, database, and the toolbox, and the evaluation strategy in verbal and spatial reasoning domains. Planning a detailed trip itinerary requires LLMs to simultaneously coordinate Points of Interest (POIs) decisions based on various preferences in the verbal reasoning dimension and optimize trip routes in spatial reasoning dimension. See Table 1 for a comparison with the previous work about newly introduced downstream tasks. In ItinBench, different levels of verbal and spatial reasoning requirements are combined into four main tasks to evaluate and compare how LLM balances between different aspects of reasoning capabilities (see Section 3.4 for detailed tasks). Given the generated itineraries in these tasks, we evaluate their failure and preference matching rate in verbal reasoning domain. More importantly, we evaluate LLMs' spatial reasoning ability through adapted Traveling Salemens Problem (TSP) (Hoffman et al., 2013) algorithm. We evaluate various models, ranging from small and large open-source models, e.g., **Llama 3.1 8B** (Dubey et al., 2024) and **Mistral Large** (Mistral, 2024), to different generations of closed-source models like **Gemini 1.5 Pro** (Team et al., 2024) and **GPT-4o** and **o1** (Hurst et al., 2024). The results indicate that LLMs struggle to maintain high and consistent planning performance when tasks from verbal and spatial reasoning domains need to be addressed simultaneously. There is only around 60% validated plan rate even when all the necessary information is

Table 1: Downstream tasks comparison between ItinBench (ours), TravelPlanner (TP) (Xie et al., 2024), ITINERA (Tang et al., 2024), and UnSatChristmas (USC) (Hao et al., 2024). RO stands for Route Optimization. Itin-Bench is the only benchmark that covers both verbal and reasoning tasks and evaluations for LLMs.

| | **Ours** | TP | ITINERA | USC |
|---|---|---|---|---|
| Preference | ✓ | ✓ | ✓ | ✓ |
| Full Open Source | ✓ | ✓ | | |
| Full Real Data | ✓ | | ✓ | ✓ |
| Day-Wise RO | ✓ | | ✓ | |
| Plan-Wise RO | ✓ | | | |
| User Review | ✓ | | | |

Table 2: Entry number for base data and their user review records. Review attributes refer to the columns in the final dataset extracted from the user reviews.

| | Base | Review Attributes |
|---|---|---|
| Restaurants | 500 | cuisine, flavor, freshness, service, environment, value |
| Hotels | 105 | quality, location, service, safety |
| Attractions | 322 | family, history, activity, nature, food, shopping |

already provided and the additionally 15% to 38% additional unnecessary travel distance in the generated plan.

Our main contributions are twofold:

- **Integrate verbal reasoning with spatial reasoning**: To better represent the complexity of real-world planning, we integrate spatial reasoning tasks and corresponding evaluations into the trip itinerary generation task to evaluate LLMs in more human-level cognitive dimensions. The downstream tasks focus on the route optimization of the POIs in the trip. We adapt TSP algorithms to enable the evaluations of various tasks in this real-world setting.

- **New evidence on LLMs' reasoning via extensive evaluations**: We quantitatively record trade-offs in LLM performance across domains when prompted for both verbal and spatial reasoning. We further find that gains on spatial tasks largely arise when models are given explicit spatial-relation cues, suggesting current "spatial reasoning" leans on semantic text manipulation rather than human-like spatial cognition.

Overall, this paper expands the evaluation of LLM planning tasks to a broader range of reasoning domains. By incorporating spatial reasoning, we create a more comprehensive testbed that reflects the complex reasoning dimensions found in real-world planning scenarios. This work broaden the dimensions in LLMs planning benchmarks instead of complicating the verbal reasoning domain.

## 2 Related Work

### 2.1 Spatial cognition

Spatial reasoning in humans is the ability to form and manipulate internal representations of space—tracking distances, directions, and relations among objects—to navigate, compare locations, and solve problems about where things are (Burgess, 2008; Byrne & Johnson-Laird, 1989). Coordinate systems act as cognitive scaffolds for spatial reasoning, letting humans encode positions, distances, and directions in egocentric or allocentric frames to compare locations and plan movement (Levinson, 2003; Herskovits, 1986). However, when our paper supplies the model with pre-computed proximity relations in text, it bypasses this spatial computation and reduce the tasks to purely semantic reasoning (Byrne & Johnson-Laird, 1989).

### 2.2 LLMs reasoning and planning

Recent work on planning with LLM agents spans commonsense task planning (Valmeekam et al., 2024; Zhao et al., 2024), tool use (Ruan et al., 2023), and pathfinding (Chen et al., 2024c; Aghzal et al., 2023); in travel domains, systems pair models with algorithmic solvers (de la Rosa et al., 2024; Ju et al., 2024), recommendation pipelines (Chen et al., 2024a), and self-correction frameworks (Xie & Zou, 2024; Hao et al., 2024; Gundawar et al., 2024), alongside benchmarks such as TravelPlanner (Xie et al., 2024), UnSatChristmas

Table 3: Preference selection details for query construction. There are six main categories, each with their option lists. Besides restaurants and hotels selecting 1 to 3 preferences with the probability weights $[0.6, 0.3, 0.1]$, all other categories choose one preference.

| Preference | Choices | Count | Probability |
|---|---|---|---|
| Day | "2 days", "3 days", "4 days" | 1 | Equal |
| Price | "cheap budget", "moderate budget", "expensive budget" | 1 | Equal |
| Attraction Orientation | "family oriented", "history oriented", "activity oriented", "nature oriented", "food oriented", "shopping oriented" | 1 | Equal |
| Restaurant Related | "good flavor", "good freshness", "good service", "good environment", "good value" | [1, 2, 3] | [0.6, 0.3, 0.1] |
| Cuisine | "US", "Mexican", "Irish", "French", "Italian", "Greek", "Indian", "Chinese", "Japanese", "Korean", "Vietnamese", "Thai", "Asian Fusion", "Middle Eastern" | 1 | Equal |
| Hotel Related | "good quality", "good location", "good service", "good safety" | [1, 2, 3] | [0.6, 0.3, 0.1] |

(Hao et al., 2024), and Triptalior (Wang et al., 2025). Yet these efforts largely probe verbal reasoning and optimize for downstream task success rather than advancing general reasoning competence.

In parallel, spatial reasoning research examines visual and spatial question answering in both LLMs and MLLMs (Yue et al., 2024; Chen et al., 2024b; Yang et al., 2025; Wang et al., 2024a), with approaches such as explicit reasoning visualization and fine-tuning to bolster performance (Wu et al., 2024b; Hu et al., 2024; Tang et al., 2025). Spatial planning work further covers path-finding (Wu et al., 2024a; Aghzal et al., 2024; Zhang et al., 2024) and route optimization with LLMs (Chen et al., 2024c; Liu et al., 2023; Fang et al., 2024). However, evaluations often rely on artificial, isolated settings (e.g., grids and board games) that underrepresent real-world conditions where multiple cognitive domains interact. Progress toward end-to-end AGI will require testbeds that integrate verbal and spatial competencies—rather than confining assessment to a single reasoning domain.

A concurrent work, TripTailor (Wang et al., 2025), focuses on personalized city-scale itinerary planning with day-level, event-specific details, whereas ItinBench provides an algorithmic evaluation of spatial reasoning by quantifying differences in final route distance.

## 2.3 Planning and reasoning in other modalities

Previous works have investigated enhancing the spatial reasoning and planning capabilities of LLMs and large multimodal models (LMMs) through curated 2D and 3D spatial reasoning datasets (Zhu et al., 2024; Ma et al., 2025) and adapted reinforcement learning strategies (Xu et al., 2025). VSI-Bench (Yang et al., 2025) evaluates video-language models in terms of spatial understanding, memory, and reasoning from video clips. PATHEVAL (Aghzal et al., 2025) positions vision-language models as plan evaluators, testing their ability to identify correct path plans. The multimodal visualization-of-thought method (Li et al., 2025) fine-tunes LMMs to interleave the generation of internal thought processes with corresponding visual representations.

PointLLM (Xu et al., 2024) explores LLMs' ability to understand and reason over 3D point clouds. The Visual Aptitude Dataset (Sharma et al., 2024) examines how string-based learning can induce latent visual and spatial understanding in LLMs. STARE (Unger et al., 2025) assesses vision-language models on spatial reasoning and manipulation tasks, such as folding and unfolding 3D objects. Different from these methods, our paper mainly focuses on evaluating the verbal and spatial reasoning ability in LLMs.

## 3 ItinBench

This section introduces each component of the ItinBench, as illustrated in Figure 1. We first present how the data pipeline and human query are constructed to enable the evaluation in the real-world setting (Section 3.1). Then, we introduce the verbal reasoning and the spatial reasoning tasks (Section 3.2) included in

the ItinBench. Additionally, we present the experiments designed (Section 3.4) and their corresponding evaluation metrics (Section 3.5).

## 3.1 Data Pipeline

To balance verbal and spatial planning tasks, Philadelphia City is chosen as an example for single-city itinerary generation. Our data contains basic information about the businesses and their reviews.

**Base Data.** The first part of the data set is the basic information about various Philadelphia businesses. Appendix 4 shows the attributes used in base data. The data is sourced from Yelp Dataset (Yelp, 2024a) and Yelp Fusion API (Yelp, 2024b). The license, intended use, and filtering is discussed in Appendix 3.3.1. It contains three main categories: restaurants, hotels, and attractions. Table 2 shows the entry number for each business category.

**User Reviews.** To better assess the reasoning capabilities of LLMs, user reviews are incorporated into the data pipeline to generate category-specific ratings, challenging the models' ability to handle detailed and precise information in rule-based setting. Table 2 shows the review number selected for each business category and the key information extracted. Appendix 3.3.1 demonstrates our review selection strategy. All user reviews for each business are compiled into separate files for key information extraction. Detailed prompts for each category are in Appendix D.1.

**Query Construction.** ItinBench provides 500 human-like queries, and each query contains various trip preferences related to the key information extracted from the user reviews. LLMs need to use these preferences as verbal reasoning clues to find the target destinations from the pools of candidates. See Table 3 for the components of all six categories of preferences. Each query incorporates 6 to 10 preferences. The generation prompt is listed in Appendix D.2.

## 3.2 Generation Pipeline

**Verbal Reasoning.** Given the nature of trip itineraries, the linguistic, logical, and temporal reasoning tasks that ItinBench offers are distinct from traditional travel planning benchmarks. The queries require LLMs to select the preferred items from a vast pool of businesses to evaluate their linguistic and deductive reasoning abilities (Figure 1). Additionally, preferences and constraints such as the trip's day length and specific attraction count further assess the LLMs' temporal and mathematical reasoning capabilities. All the tasks are from the verbal reasoning domain but from a different approach comparing to previous travel planning tasks.

**Spatial Reasoning.** In ItinBench, LLMs' spatial reasoning ability is evaluated through various route optimization tasks. These tasks require LLMs to independently infer and visualize spatial relationships among the POIs. The goal is to minimize unnecessary travel distance based on attractions' addresses, latitude, and longitude information. In specific tasks, we provide the LLMs with the spatial cluster information of the attractions and hotel candidates in text. Appendix 3.3.2 details how the clusters are calculated. This aims to better determine which reasoning ability LLMs use when performing spatial reasoning tasks. The hypothesis – LLMs rely on using their verbal reasoning abilities to draw connections between text data and their training knowledge to perform spatial reasoning tasks – is introduced and discussed in Section 4.2 and a case study in Section 4.5.

## 3.3 Dataset Details

### 3.3.1 Data Pre-process Strategy

The base dataset is collected from Yelp Dataset (Yelp, 2024a). The dataset is a subset of the Yelp Dataset that intended for educational use. The original dataset contains the user reviews that are hard to monitor in full scale, thus the final dataset provided only contains the rating not the raw review manuscript. The dataset is in English fully.The price attributes are obtained from Yelp Fusion API (Yelp, 2024b).

Table 4: The attributes for basic data

| Preference | Attributes |
|---|---|
| Hotels | "business_id", "name", "address", "latitude", "longitude", "stars", "price" |
| Restaurants | "business_id", "name", "address", "latitude", "longitude", "stars", "price", "good_for_meal", "cuisine_1", "cuisine_2" |
| Attraction | "business_id", "name", "address", "latitude", "longitude", "stars", "price" |

**Hotels** We extract businesses with the "Hotels" category only from the original data since the category "Hotel & Travel" actually refers to airports or train stations.

**Restaurants** We extract businesses with "Restaurants" or "Food" categories from the original data. We keep the top 500 restaurants with the most reviews for efficiency and cost management.

**Attractions** We extract businesses with "Museums", "Parks", "Local Flavor", "Zoos", "Tours", "Landmarks & Historical Buildings", and "Souvenir Shops" categories from the original dataset.

**User Reviews** We keep reviews with a "useful" rating greater or equal to 1 from the original dataset. The pre-process can help filter out less informative reviews and control the file size.

All businesses labeled as unopened are filtered, and further standard data cleaning is performed. See Table 4 for the attributes we keep for each category for the base data.

### 3.3.2 Spatial Cluster Information Generation

Spatial cluster information is calculated and presented to LLMs in two ways.

**Filtered Data** In this task, business data for each plan is already filtered based on the preference requested from the query. The k-mean clustering method is deployed to get the candidates' spatial clustering information. The cluster number we choose is the integer value after using the candidate's number divided by 5.

**Tool use** In tool use tasks, LLMs will gather candidates' business through their tool calling, and then spatial clustering information based on those candidates will be provided when they successfully call the clustering function. The clustering strategy is the same as the filtered data tasks.

### 3.4 Design of Experiments

Four tasks with three different greedy approaches are proposed. All the experiments follows the API call style thus doesn't requires GPU memory to perform the inference. All temperatures are set to 1, except for OpenAI o1, which is set to 1. We collect the output for a single run.

- **Greedy Algorithm.** A greedy algorithm provides an interpretable heuristic that serves as a baseline benchmark. It uses a filtering method then heuristically arrange the filtered POIs using a minimum-distance strategy. Additional planning algorithm variants are discussed in Appendix A.1.

- **All Data with No Route Optimization.** The first task uses the entire dataset for LLMs to arrange the trip itinerary without any requests for route optimization. See Appendix D.3 for a detailed prompt. This task challenges LLMs' verbal reasoning abilities, e.g., semantic comprehension and inference reasoning abilities, in isolation, free from the influence of spatial reasoning demands.

- **All Data with Route Optimization.** The second task uses the entire dataset but introduces the requests for route optimization. See Appendix D.4 for a detailed prompt. This task evaluates LLMs' multi-task reasoning performance when handling complex verbal tasks while simultaneously addressing spatial optimization requests.

- **Filtered Data with Route Optimization.** The third task provides the data already filtered based on preferences mentioned in the query while still requiring route optimization. See Appendix

D.4 for detailed prompt. With significantly less verbal reasoning challenge, this task evaluates LLMs' planning performance when their primary focus shifts to solving spatial reasoning tasks.

- **Tool Use with Route Optimization.** The fourth task adds a new reasoning and planning dimension, i.e., tool use. LLMs are provided with a React-style (Yao et al., 2022) prompt and a set of custom tools listed in Appendix D.5. They need to identify and call the tools appropriately to gather information during the inference. This additional tool-use task enables evaluating LLMs' reasoning and planning ability through a more real-world-like scenario.

### 3.5 Evaluation

After the generation, the key POIs information is extracted by LLMs, similar to other related works (Xie et al., 2024; Hao et al., 2024). The extraction prompt is in Appendix D.6.

#### 3.5.1 Verbal Reasoning

We first evaluate LLM planning performance through failure checks, then adopt **Micro** and **Macro** calculations from TravelPlanner (Xie et al., 2024), Finally, the **Validated Rate (VR)** measures the proportion of plans that successfully pass all the failure checks and preference checks.

**Out of Pool (OOP).** Since LLMs learn world knowledge during their training phase (Huang et al., 2023), they might provide choices that appear in the training dataset but not in the given data. OOP is calculated as:

$$\text{Out of Pool} = \frac{\sum_{p \in P} \mathbb{1}_{O(p)}}{|P|}, \tag{1}$$

where $P$ stands for the plans that are evaluated, and $O(p)$ is a function that determines if plan $p$ contains at least one piece of out-of-pool information.

**Missing Information (MI).** LLMs might provide vague recommendations, e.g., "Wandering around the south city," or fail to provide any information for certain planned activities. The MI rate is calculated as:

$$\text{Missing Information} = \frac{\sum_{p \in P} \mathbb{1}_{M(p)}}{|P|}, \tag{2}$$

where $M(p)$ is a function that determines if a plan $p$ contains at least 1 missing information entries.

**Micro.** Given a set of preferences from the human query, each related recommendation in the itinerary incurs a new entry for evaluation. Thus, the micro rate measures the percentage of the entries in their itineraries that satisfied their corresponding preferences. Entries in the plans that fail the failure check won't be further evaluated. Micro rate is calculated as:

$$\text{Micro} = \frac{\sum_{p \in P} \sum_{q \in Q_p} \sum_{e \in E_{pq}} \mathbb{1}_{passed_1(e,q,p)}}{\sum_{p \in P} \sum_{q \in Q_p} |E_{pq}|}, \tag{3}$$

where $Q_p$ stands for the sets of preferences that apply to a plan $p$, $E_{pq}$ stands for sets of entries in a plan $p$ related to their preferences $q$. $passed_1(e, q, p)$ stands for a function determine if an entry $e$ in a plan $p$ regarding its related preference $q$ is satisfied.

**Macro.** The macro rate measures the percentage of the plans whose micro rate is higher than a predefined threshold. In ItinBench, the threshold is set to 75%. This is a flexible threshold and a 75% threshold allows up to one unmet preference out of four in each evaluation category. Plans containing missing information or out-of-pool choices are not excluded from the evaluation, but the specific entries are skipped while calculating the macro rate. The macro rate is calculated as:

$$\text{Macro} = \frac{\sum_{p \in P} \mathbb{1}_{passed_2(Q_p, E_{pq}, \alpha)}}{|P|}, \tag{4}$$

where $passed_2(Q_p, E_{pq}, \alpha)$ is a function determine if all the entries $E_{pq}$ for plan $p$ satisfies the set of preferences $Q_p$ with a percentage greater than the threshold $\alpha$.

**Validated Rate (VR).** The validated rate measures the percentage of the plans that pass the failure check and the threshold set in the macro calculation. It serves as the overall evaluation of the verbal reasoning domain. The validated rate is calculated as:

$$\text{VR} = \frac{\sum_{p \in P} \mathbb{1}_{M'(p)} \mathbb{1}_{O'(p)} \mathbb{1}_{passed_2(Q_p, E_{pq}, \alpha)}}{|P|}, \tag{5}$$

where $M'(p)$ is a function determine if a plan $p$ contains no missing entry. $O'(p)$ is a function determine if a plan $p$ contains no out-of-pool entry.

### 3.5.2 Spatial Reasoning

**Average Recommendation Gap (ARG).** We require LLMs to propose exactly four attractions daily for fairness and consistency across tasks. The average recommendation gap measures the deviation of the number of attractions recommended in the generated itinerary from this 4-attraction requirement and is calculated as:

$$\text{ARG} = \frac{\sum_{p \in P} \sum_{d \in D_p} (|A_{pd}| - \beta)}{\sum_{p \in P} |D_p|}, \tag{6}$$

where $D_p$ stands for a set of days in a plan $p$, $A_{pd}$ stands for a set of daily attractions recommended in a plan $p$ at day $d$, and $\beta$ stands for the number of daily plans requested by us.

**Distance Gap (DG).** Day-wise arrangement is a classical Traveling Salesmen Problem (Hoffman et al., 2013). The distance gap measures the distance difference daily between the optimized and LLM-proposed routes. Days that fail the failure checks are excluded from this evaluation. The distance gap is calculated as:

$$\text{DG} = \frac{\sum_{p \in P} \sum_{d \in D_p} (C(A_{pd}, H_{pd}) - C'(A_{pd}, H_{pd}))}{\sum_{p \in P} |D_p|}, \tag{7}$$

where $H_{dp}$ stands for hotels proposed by a plan $p$ at day $d$, $C(X, Y)$ stands for the calculated distance by the LLM generated plan, and $C'(X, Y)$ is the calculated distance by the optimized plan.

**Total Distance Gap (Total-DG).** For plan-wise evaluation, the distance gap measures the difference in travel distance between the optimized and the LLM proposed route for the entire plan. Our adapted TSP algorithm enables evaluations for multiple hotel choices and cities throughout the trip. The algorithm is detailed in Appendix E.1. The total distance gap is calculated as:

$$\text{Total-DG} = \frac{\sum_{p \in P} (C(A_p, H_p) - C'(A_p, H_p))}{|P|}. \tag{8}$$

**Extra Cluster Jump (ECJ).** The extra cluster jump evaluates how well LLMs visualize and understand spatial relationships among attractions. An optimized clustering strategy among attractions and hotels theoretically exists for each itinerary. The extra cluster jump measures the number of times the LLM proposed route deviates from this strategy by visiting attractions that is from further clusters instead of choosing nearby attractions within the same cluster as the current day, which is calculated as:

$$\text{ECJ} = \frac{\sum_{p \in P} (N_p - N'_p)}{\sum_{p \in P} N'_P}, \tag{9}$$

where $N'_p$ stands for the number of clusters calculated by the optimized clustering strategy, and $N_p$ stands for the number of clusters visited by the generated plan based on this strategy.

Table 5: In percentage, the evaluation of LLM's trip itinerary generation in different tasks is presented, with the best results highlighted in bold. The validated plan is around 7% and 65% with or without the filtered data accessible to the LLMs. When ask the LLM to perform the route optimization, the total distance gap is still around 20% for older models and 7% for newer models like o1 with and without the access to the spatial clustering information. Gemini's spatial reasoning task result is "-" since it failed to provide a rational number of attractions.

| | Verbal Reasoning | | | | | Spatial Reasoning | | | |
| --- | --- | --- | --- | --- | --- | --- | --- | --- | --- |
| | OOP ↓ | MI ↓ | Micro ↑ | Macro ↑ | VR ↑ | ARG ↓ | DG ↓ | Total-DG ↓ | ECJ ↓ |
| Greedy Approach | 0.0 | 0.0 | 100.0 | 100.0 | 100.0 | 0 (4.00) | 4.0 | 9.2 | 86.2 |
| Task 1: Entire Dataset, No Request For Route Optimization (#100) | | | | | | | | | |
| Llama 3.1 8B | 45.0 | 11.0 | 60.1 | 0.0 | 0.0 | 24.3 (3.03) | 9.2 | 24.6 | **99.2** |
| Mistral-large (123B) | 52.0 | 0.0 | 66.9 | 2.0 | 2.0 | 1.2 (3.95) | **6.5** | 27.9 | 113.1 |
| Gemini-1.5-Pro | 18.0 | 52.0 | 77.0 | 5.0 | 5.0 | 13 (3.48) | 7.7 | 25.2 | 124.3 |
| GPT-4o-2024-11-20 | 13.0 | 0.0 | 77.3 | 5.0 | 5.0 | 1.3 (3.95) | 12.1 | 38.0 | 146.3 |
| OpenAI o1 | **6.0** | **0.0** | **86.2** | **20.0** | **18.0** | **0.75 (3.97)** | 9.2 | **24.0** | 128.2 |
| Task 2: Entire Dataset, Request For Route Optimization (#100) | | | | | | | | | |
| Llama 3.1 8B | 51.0 | 12.0 | 59.7 | 0.0 | 0.0 | 25.6 (2.98) | 7.2 | 24.9 | 104.8 |
| Mistral-large (123B) | 52.0 | 0.0 | 68.4 | 0.0 | 0.0 | 0.2 (4.01) | 6.8 | 26.9 | 115.9 |
| Gemini-1.5-Pro | 23.0 | 11.0 | 77.6 | 8.0 | **7.0** | 25 (5.0) | - | - | - |
| GPT-4o-2024-11-20 | 20.0 | **0.0** | 76.1 | 4.0 | 4.0 | 1.8 (3.93) | 11.1 | 28.5 | 127.0 |
| OpenAI o1 | **12.0** | 12.0 | **81.9** | 4.0 | 4.0 | **1.3 (3.95)** | **6.2** | **9.1** | **49.0** |
| Task 3: Filtered Dataset, Request For Route Optimization (#300) | | | | | | | | | |
| Llama 3.1 8B | 20.7 | 23.0 | 91.2 | 66.3 | 51.0 | 10.0 (3.60) | 8.0 | 23.2 | 115.7 |
| Mistral-large (123B) | **11.0** | **0.0** | **95.6** | **69.7** | **66.7** | 1.0 (4.04) | 7.3 | 16.8 | 104.3 |
| Gemini-1.5-Pro | 30.0 | 28.0 | 80.6 | 20.0 | 15.0 | 22.8 (4.91) | - | - | - |
| GPT-4o-2024-11-20 | 28.0 | 0.0 | 93.1 | 56.0 | 52.5 | 0.2 (3.99) | 7.5 | 15.2 | 52.7 |
| OpenAI o1 | 30.0 | 8.0 | 89.5 | 42.0 | 42.0 | **0.2 (4.01)** | **7.3** | **7.5** | **6.9** |
| Task 4: Tool Use, Request For Route Optimization (#300) | | | | | | | | | |
| Llama 3.1 8B | 38.0 | 15.3 | 83.4 | 37.0 | 26.3 | 10.3 (4.41) | 10.0 | 27.5 | 128.0 |
| Mistral-large (123B) | **13.0** | **0.3** | **95.2** | **69.3** | **64.0** | **0.0 (4.00)** | **6.4** | 18.1 | 112.5 |
| Gemini-1.5-Pro | 24.0 | 47.0 | 79.7 | 23.0 | 16.0 | 29.0 (5.16) | - | - | - |
| GPT-4o-2024-11-20 | 20.7 | 2.7 | 93.1 | 60.0 | 57.0 | 0.2 (3.99) | 7.0 | 14.7 | 65.8 |
| OpenAI o1 | 27.0 | 2.0 | 89.4 | 41.3 | 42.3 | 0.5 (4.02) | 7.3 | **7.7** | **7.2** |

## 4 Main Results

### 4.1 Verbal Reasoning

ItinBench presents a verbal reasoning challenge for LLMs. As shown in Table 5, LLMs produce plans with up to 51% out-of-pool selections and up to 52% missing information. Even when given access to the entire dataset, the highest validated plan rate is only 18%, achieved by o1. However, when models are provided with pre-filtered data—aligned with preferences specified in the user query—the validated plan rate increases significantly, reaching up to 66.7%. Figure 2 illustrates the performance differences across tasks under various data access conditions. This substantial improvement highlights that LLMs currently struggle with detailed reasoning in multi-rule, multi-step settings, particularly when required to independently identify and apply relevant constraints.

### 4.2 Spatial Reasoning

**LLMs rely on additional textual cues to improve spatial reasoning results.** There is no significant improvement in spatial reasoning performance when models are explicitly instructed to optimize routes unless clustering information is provided in textual form, which reduces the problem to a semantic level and bypasses genuine spatial reasoning. As shown in Figure 2b, the primary difference between Task 1 and Task 2 is that Task 2 requests route optimization without providing additional spatial information; however, the Total Distance Gap (Total-DG) does not decrease. In contrast, when spatial clustering information is introduced, the Total-DG is reduced from approximately 25% to 15%. The Extra Cluster Jump (ECJ)

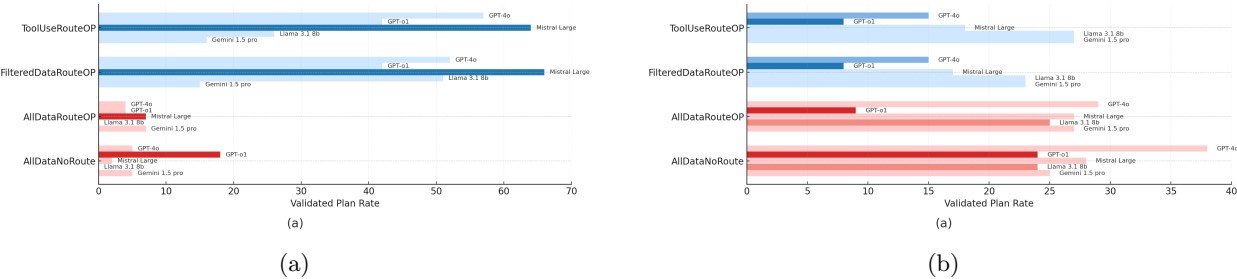

(a)                  (b)

Figure 2: Visualizations of the main results for Validated Rate (VR) and Total Distance Gap (Total-DG). Task 1 and Task 2 (red) do not have access to filtered data. Task 3 and Task 4 (blue) have access to filtered data and spatial clustering information. The second-best result is shown in darker color, and the best result is shown in the darkest color.

metric also decreases substantially—from over 100% to around 50%. This phenomenon is further illustrated in the case study in Appendix 4.5.

**All models show performance trade-offs when facing dual-domain reasoning tasks.** The newer reasoning model o1 achieves approximately 7% to 9% Total-DG when explicitly prompted to optimize routes. However, compared to its earlier 10% lead in validated rate (VR), its verbal reasoning performance declines when spatial optimization is emphasized, resulting in performance approximately 20% behind the leading model in verbal reasoning tasks. Other models exhibit similar trade-offs to varying degrees.

### 4.3 Tool Use

As shown in Table 6, GPT-4o and Mistral Large achieve 100% delivery rates in the tool-use task. Llama 3.1 8B and Gemini 1.5 Pro achieve delivery rates of 84.7% and 72%, respectively. The parameter accuracy for all models exceeds 60%. For GPT-4o, most parameter errors arise from hotel- and restaurant-related tool calls, as shown in Figure 3. Preferences associated with these categories often appear diffusely across the query, making them harder to identify than explicit constraints such as budget or trip duration. These results indicate that LLMs' verbal reasoning abilities—particularly fine-grained semantic understanding—remain challenged by complex, detail-heavy user queries.

|  | Llama | Mistral | Gemini | GPT |
|---|---|---|---|---|
| Parameter ACC | 58.4 | 62.6 | 61.3 | **62.9** |
| Delivery Rate | 84.7 | 100 | 72.0 | 100 |
| Order Dead Loop | 48.9 | – | 0.0 | – |
| Argument DL | 51.1 | – | 100.0 | – |

Table 6: Tool-use performance for Llama 3.1 8B, Mistral Large, Gemini 1.5 Pro, and GPT-4o. DL denotes dead loop.

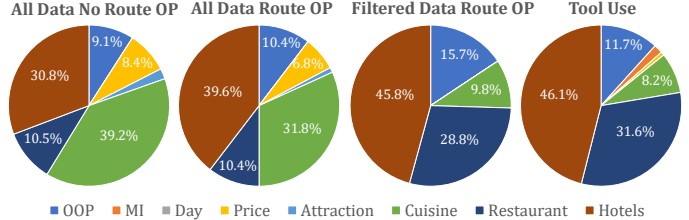

Figure 3: Error distribution for GPT-4o across four tasks. Errors primarily occur in out-of-pool, cuisine, restaurant, and hotel-related recommendations.

### 4.4 Human Evaluation

### 4.4.1 Evaluation Setup

We conduct two human evaluation tasks with 12 volunteers (10 PhD students and 2 experienced travelers), resulting in 120 judgments per task. In both tasks, annotators compare four plans for the same query, where each plan includes natural language text and a route visualization.

**Evaluation 1** compares plans generated by four different LLMs under the Filtered Data Route OP setting. **Evaluation 2** compares plans generated by OpenAI-o1 under four different task settings.

### 4.4.2 Human Evaluation Results

Human preferences are broadly consistent with the automatic metrics in ItinBench. In Evaluation 1, GPT-4o and Mistral receive the highest preference rates, while Gemini and Llama rank lower. In Evaluation 2, FilteredData and ToolUse are preferred more often than the other settings. Figure 4 shows that human preferences generally follow the same trends as validation rate and inverted Total-DG, suggesting that the benchmark is reasonably aligned with human judgment. Quantitative result is recorded in Appendix B

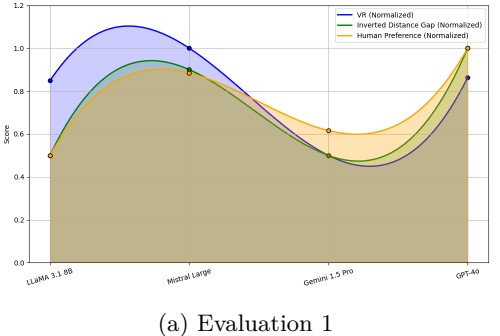

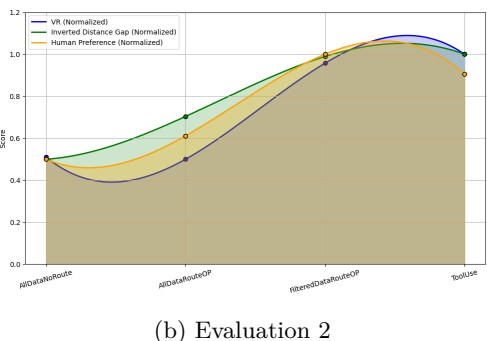

(a) Evaluation 1                                        (b) Evaluation 2

Figure 4: Alignment between human preferences, validation rate, and inverted Total-DG.

### 4.5 Case Study

In Figure 5, we visualize an itinerary generated by GPT-4o for the spatial reasoning task in the tool-use mode. The green circles in Figure 5c are drawn based on the clustering information provided in Figure 5b, with Day 1's route highlighted in red. GPT-4o successfully captures part of the spatial structure of the problem by grouping attractions from the same cluster into the same day. This suggests that the model can make effective use of the cluster assignments and preserve local coherence when constructing a daily itinerary. At this level, the generated plan appears spatially plausible, since nearby attractions are generally kept together rather than being arbitrarily distributed across different days.

However, a closer comparison between Figure 5c, which shows the proposed route, and Figure 5d, which shows the optimized route, reveals an important flaw. The fourth attraction selected for Day 1 is spatially unreasonable: it belongs to a cluster that is substantially farther from the other clusters assigned to Day 1, while being much closer to the group of clusters scheduled for Day 2. As a result, although the itinerary looks coherent at the attraction level, it becomes suboptimal when examined at the higher route-planning level. This example shows that producing a locally reasonable plan does not necessarily imply globally consistent spatial organization.

From a spatial reasoning perspective, arranging one day in an itinerary involves two related but distinct tasks. The first is to identify spatial relationships among individual attractions and use them to form first-level clusters. The second is to reason over the spatial relationships among these clusters and organize them into an efficient route for the day as a whole. Although these tasks operate at different granularities, they share the same underlying requirement: understanding relative positions and distance proximity in a two-dimensional space. In other words, the second task can be viewed as a higher-level extension of the first, where clusters become the objects of reasoning instead of individual attractions.

This case study suggests that GPT-4o performs reasonably well on the first task but struggles on the second. In particular, it fails to recognize that first-level cluster 13 is far from clusters 17 and 8, which are also assigned to Day 1. That failure indicates a limitation in the model's ability to form a consistent global spatial representation beyond immediately given grouping information. More broadly, the result implies that the model's apparent success in selecting accurate first-level clusters may rely less on genuine spatial

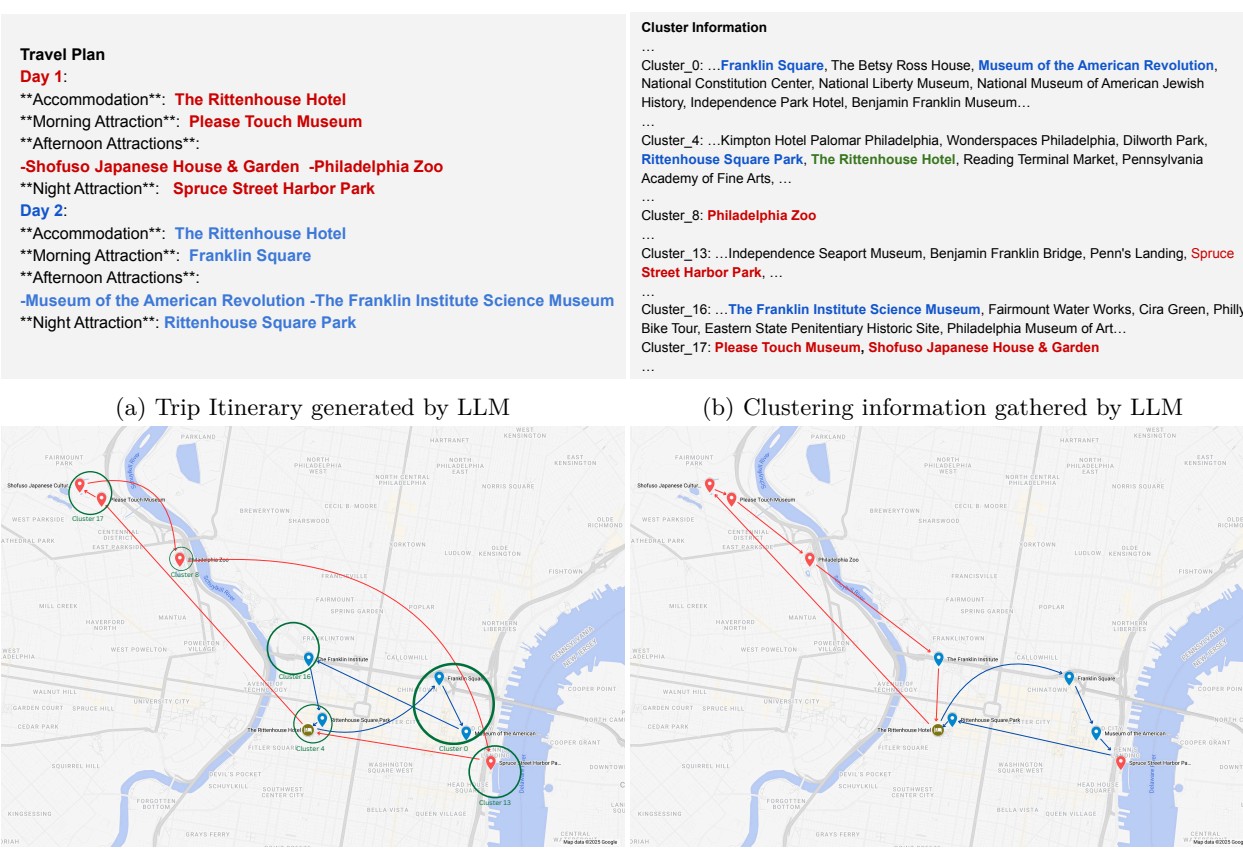

(a) Trip Itinerary generated by LLM

(b) Clustering information gathered by LLM

(c) The generated route. Red: Day 1; Blue: Day 2; Green: Clusters

(d) Optimized route calculated by the adapted TSP algorithm

Figure 5: Visualization of a case study about the itinerary generated by GPT 4o in tool-use mode, drawn in Google Map (GoogleMaps, 2025). Plan-wise (red markers and routes in Figure 5c), one of the main issues is the route leads to the bottom right corner of the map (Cluster 13) while visiting the same area (Cluster 0) on the second day again. The mistake is corrected in the optimized route in Figure 5d. For this itinerary, the total distance gap ratio is 25.6%. Additionally, the extra cluster jump ratio is 100%.

reasoning and more on verbal reasoning over the provided clustering information. Thus, while GPT-4o can exploit explicit symbolic cues effectively, its ability to truly visualize and reason about spatial relations in the underlying map remains limited.

# 5 Limitations and Future Work

Empirically and in prior work, most "spatial" gains arise when spatial information is rendered as text—via fine-tuning, tool outputs, or multimodality with textual descriptions—rather than from improved geometric computation (Wang et al., 2024a; Tang et al., 2025; Wu et al., 2024b; Chen et al., 2024b). This raises a key question: are we fostering human-like spatial cognition or optimizing with propositional shortcuts?

Future work should reduce reliance on such shortcuts in evaluation, explore models that operate natively over structured spatial representations (e.g., maps, graphs, and coordinates), and extend ItinBench to multi-city and variable-constraint settings to test for genuine spatial computation rather than semantic cue amplification. This work serves as an initial test demonstrating that introducing additional cognitive tasks can challenge LLMs.

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

# A More experiments and results

## A.1 Greedy Algorithm

- $A^*$ Algorithm Employs a Minimum Spanning Tree as the heuristic h(n).

- Mixed Integer Algorithm Uses continuous ordering variables in the Miller–Tucker–Zemlin (MTZ) subtour elimination constraints.

Table A.1: Alternative baselines; columns correspond to Table 5 in the main text. Except for the greedy + min distance approach, all other approaches achieve perfect performance in evaluation.

| | Verbal Reasoning | | | | | Spatial Reasoning | | | |
| --- | --- | --- | --- | --- | --- | --- | --- | --- | --- |
| | OOP ↓ | MI ↓ | Micro ↑ | Macro ↑ | VR ↑ | ARG ↓ | DG ↓ | Total-DG ↓ | ECJ ↓ |
| Greedy Approach (#100) | | | | | | | | | |
| $A^*$ | 0.0 | 0.0 | 100.0 | 100.0 | 100.0 | 0 (4.00) | 0.0 | 0.0 | 0.0 |
| MIP | 0.0 | 0.0 | 100.0 | 100.0 | 100.0 | 0 (4.00) | 0.0 | 0.0 | 0.0 |

The near-perfect performance of algorithm-based solutions aligns with findings in combinatorial optimization research. These alternative baselines strongly indicate that LLMs' reasoning and planning abilities still require improvement. Notably, these performances depend on extensive preparation, task-specific code, and prior human knowledge where LLMs can facilitate end-to-end reasoning solutions.

# B Human Evaluation

## B.1 Quantitative Results

Across the two human evaluation tasks, the preference rates are broadly consistent with the automatic measurements defined in ItinBench. In Evaluation 1, human judgments follow the same overall ranking pattern observed in both validation rate (VR) and the inverted plan-wise distance gap (Total-DG), indicating that models with stronger benchmark performance are also more likely to be preferred by annotators (Figure 4a). This agreement is visible not only in the general trend of the curves, but also in the relative ordering shown in Table B.2: GPT-4o receives the highest human preference rate, followed by Mistral, while Gemini and Llama are ranked lower. Although the three curves are not identical at every point, they move in a similar direction across models, suggesting that the benchmark metrics capture meaningful aspects of plan quality that are also recognized by humans.

A similar pattern appears in Evaluation 2, where human preferences are compared across different task settings using plans generated by OpenAI-o1. As shown in Figure 4b, the human preference curve again tracks the trends of VR and inverted Total-DG, with stronger task settings under the benchmark also receiving higher human preference. The quantitative results in Table B.3 further support this observation: FilteredDataRouteOP achieves the highest preference rate, ToolUse follows closely behind, and the remaining settings receive noticeably lower scores. Taken together, these results provide additional evidence that ItinBench is well aligned with human judgment at both the cross-model level and the cross-setting level.

# C Reproducibility

We attach the code used to generate the results and conduct evaluation in the supplementary material. We plan to open-source the code and dataset in the future. The data crafting pipeline is detailed in Section 3.1.

Table B.2: Preferred rate across the plans generated by four different LLMs. Evaluation 1 refers to the comparison between the plans generated by four LLMs based on the same user query.

|  | Mistral | GPT4o | Llama | Gemini |
| --- | --- | --- | --- | --- |
| Evaluation 1 | 30.8 | 35.8 | 14.2 | 19.2 |

Table B.3: Preferred rate across the plans generated under different task settings. Evaluation 2 refers to the comparison between plans generated by OpenAI-o1 under different task settings for the same user query.

|  | AllDataNoRoute | AllDataRouteOP | FilteredData | ToolUse |
| --- | --- | --- | --- | --- |
| Evaluation 2 | 20.5 | 19.8 | 33.2 | 30.5 |

# D  Prompts

## D.1  Review Extraction Prompt

There is one prompt for each of the business categories. The prompt asks LLM to extract ratings or measurements on different scales. These numbers are processed into phrases, e.g., rating 5 for location is "excellent location," for planner LLM to better understand.

Here is the prompt for hotel review extraction.

> You are an assistant designed to summarize reviews of businesses for travel planning purposes. Your goal is to provide **faithful, concise, and relevant information** based on the following reviews complied into the txt file. Follow these principles:
>
> 1. **Focus on Travel-Relevant Details:** Prioritize aspects like location convenience, proximity to landmarks, transportation options, ambiance, cleanliness, service quality, amenities, and overall reliability.
>
> 2. **Avoid Bias:** Reflect the consensus of reviews, clearly noting if opinions are mixed. Do not add, fabricate, or exaggerate details.
>
> 3. **Clarify Nuances:** Mention trends (e.g., "frequent mentions of slow service" or "consistent praise for central location").
>
> 4. **Respect Context:** Differentiate between subjective opinions (e.g., "some reviewers found the rooms small") and factual details (e.g., "located 5 minutes from the train station").
>
> 5. **Stay Honest:** If the reviews are unclear or contradictory, state this explicitly rather than drawing unsupported conclusions.
>
> 6. **Highlight Red Flags or Unique Strengths:** Identify issues (e.g., safety concerns, unexpected fees) or advantages (e.g., exceptional customer service, standout features).
>
> Output formatting instructions:
>
> On a scale of 1 to 5. 3 means average, 4 means good, 5 means excellent, 2 means below average, and 1 means bad. Be faithful and give objective ratings.
>
> 1. Evaluate Room Quality on a scale from 1 to 5. Considering size, cleanliness, space, amenities, noise level, and other considerations.
>
> 2. Evaluate the location and convenience on a scale from 1 to 5. Consider transportation options, proximity to attractions, and other factors.
>
> 3. Evaluate the hotel's service on a scale from 1 to 5, considering the cleaning service, customer service, valet service, check-in and check-out experience, and interactions between travelers and the hotel staff in general.
>
> 4. Evaluate the safety on a scale from 1 to 5. Considering the surrounding area traffic, safety in the hotel, and other factors that influence the safety concern if possible.

Give one evaluation for each attribute and followed by a sentence of reasoning.

—— Example 1 Starts ——

The hotel has a rating of 4 for quality. Rooms are beautifully appointed with stunning views, luxurious amenities, and impeccable cleanliness. Guests appreciate the spaciousness and comfort of the beds, although some mention the rooms being on the smaller side typical for city hotels.

The hotel has a rating of 5 for location. Located in the Comcast Center, the hotel offers breathtaking views of Philadelphia and is conveniently situated near major attractions. The elevator ride to the 60th floor lobby is a highlight.

The hotel has a rating of 4 for service. Service is generally exceptional, with staff going above and beyond to make guests feel welcome. However, there are mixed reviews regarding the handling of certain situations, particularly in the bar area and restaurant.

The hotel has a rating of 4 for safety. The hotel is located in a prominent area of Philadelphia, and while most reviews do not raise safety concerns, there are mentions of discriminatory treatment that could affect the perception of safety for some guests.

—— Example 1 Ends ——

—— Example 2 Starts ——

The hotel has a rating of 2 for quality. Rooms are often reported as dirty, with issues like stained bedding, bugs, and unclean bathrooms. Some guests noted that while the rooms are spacious, they are poorly maintained and have unpleasant odors.

The hotel has a rating of 3 for location. The hotel is conveniently located near the airport, but guests noted that the surrounding area lacks amenities and attractions, requiring a drive for most necessities.

The hotel has a rating of 2 for average service. Service quality is inconsistent, with many guests reporting rude or unhelpful staff. Issues with check-in, maintenance, and customer service have been frequently mentioned.

The hotel has a rating of 2 for average safety. Concerns about safety have been raised, particularly regarding the external room entrances and reports of security issues. Some guests felt uncomfortable due to the behavior of staff and security.

—— Example 2 Ends ——

Given reviews: {reviews}

Your evaluation:

Here is the attraction review extraction prompt:

You are an assistant designed to analyze and summarize reviews of attractions for travel planning purposes. Your goal is to deliver faithful, concise, and travel-relevant insights based on the reviews provided in the attached text file. Follow these principles:

1. Focus on Key Travel-Relevant Features: Highlight details such as the attraction's location, accessibility, proximity to key landmarks, transportation options, and overall convenience for visitors. Address aspects like ambiance, cleanliness, crowd levels, staff behavior, unique offerings, and amenities.

2. Reflect Consensus and Avoid Bias: Summarize the general sentiment of reviewers, noting both strengths and shortcomings as expressed. Avoid exaggeration or unfounded interpretations. Indicate if opinions vary significantly among reviewers. Clarify Trends and Nuances:

3. Identify recurring themes (e.g., "many reviewers appreciated the tranquil setting" or "frequent complaints about high entrance fees"). Distinguish between subjective opinions (e.g., "some visitors found it too crowded") and objective facts (e.g., "located 10 minutes from the nearest metro station"). Acknowledge Uncertainty or Contradictions:

4. If reviews are unclear or contradictory, explicitly state this rather than making unsupported conclusions.

5. Highlight Red Flags or Unique Features: Draw attention to notable issues (e.g., safety concerns, hidden costs) or standout positives (e.g., spectacular views, interactive exhibits).

Output formatting instructions: All the evaluation is on a scale of 0 to 3, 0 means not applicable, 1 means low tendency, 2 means medium, and 3 means strong tendency. The scale is not a score but a measurement. There is no implication that a better score leads to a better business.

1. Measure the family orientation from 0 to 3. Factors include kids involvement, and What kinds of activities are organized? 0 means not for family, 1 means really small family factor is designed, 2 means an average amount of family activities, and 3 means this place designed for family.

2. Measure the history oritentaion from 0 to 3. Factors include history, culture, education, and other considerations around history and culture. 0 means no history consideration from this site, 1 means not designed for history exploration, 2 mean average amount of history attributes, 3 means this place has a lot of history factor included.

3. Measure the activity level from 0 to 3. This measures what level of action is needed for this attraction. Hiking or dangerous activities would be a strong activity level 3, visiting a outdoor park could be a medium level 2, and visiting a museum could be a low activity level 1.

4. Measure the natural scene from 0 to 3. This measures how much the attraction accesses nature and sightseeing views. 0 means completely indoor, and 3 means outdoor with the natural scene.

5. Measure how food-oriented is the attraction. Level 3 would be food oriented attraction. 0 indicates this attraction has no relation to food.

6. Measure if attraction focus on shopping. A market would be level 3, a historical landmark could be 0 since it's for visiting only.

Here are some examples

—— Example 1 starts ——

This place has a family oriented level 3. Many families enjoyed the carriage rides, with children actively participating and asking questions. The experience was highlighted as a memorable family activity.

This place has a history oriented level 3. The carriage rides provide informative tours of historical areas, with knowledgeable guides sharing insights about Philadelphia's history and architecture.

This place has a activity oriented level 1. The activity level is low as the rides are leisurely and do not require physical exertion from participants.

This place has a nature oriented level 1. The rides are primarily through urban areas with limited access to natural scenery, focusing more on the city's historical aspects.

This place has a food oriented level 0. The attraction does not have a food-related focus.

This place has a shopping oriented level 0. The carriage rides are not related to shopping; they are purely a sightseeing experience.

—— Example 1 Ends ——

—— Example 2 Starts ——

This place has a family oriented level 3. Spruce Street Harbor Park is highly family-friendly, featuring activities for children such as oversized games, an arcade, and play areas. Many reviewers noted the park's appeal to families, with fun events and games for kids.

This place has a history oriented level 1. While the park is located near historical sites, it does not focus on history or cultural education. The attraction is more about leisure and entertainment rather than historical significance.

This place has an activity oriented level 2. The park offers various activities such as hammocks, games like giant Jenga and Connect Four, and paddle boat rentals. However, the level of physical activity is moderate, making it suitable for casual visitors.

This place has a nature oriented level 2. The park is situated along the Delaware River and features hammocks and seating areas with views of the water. However, it is primarily an urban park with limited natural scenery.

This place has a food oriented level 3. There is a strong focus on food, with numerous food trucks and vendors offering a variety of options, including local favorites. Reviewers praised the food offerings, although some noted that prices can be high.

This place has a shopping oriented level 1. While there are some vendors selling crafts and local goods, shopping is not a primary focus of the park. The main attractions are food and recreational activities.

—— Example 2 Ends ——

Given reviews: {reviews}

Your evaluation:

Here is the restaurant review extraction prompt:

You are an assistant designed to summarize reviews of businesses for travel planning purposes. Your goal is to provide **faithful, concise, and relevant information** based on the following reviews complied into the txt file. Follow these principles:

1. Focus on Travel-Relevant Details: Prioritize aspects crucial to travelers, such as food quality, location convenience (proximity to landmarks and transportation options), ambiance, cleanliness, service quality, amenities, and overall reliability.

2. Avoid Bias: Provide balanced evaluations that reflect the consensus of available reviews. Clearly indicate when opinions are mixed, and refrain from fabricating, exaggerating, or omitting key details.

3. Clarify Nuances: Highlight notable trends in feedback (e.g., "frequent mentions of slow service" or "consistent praise for convenient location") to provide an accurate overview.

4. Respect Context: Differentiate between subjective opinions (e.g., "some diners found the portions small") and factual details (e.g., "located within walking distance of a major metro station").

5. Maintain Honesty: If reviews are unclear, contradictory, or lacking sufficient detail, explicitly state this instead of making unsupported conclusions.

6. Highlight Red Flags and Unique Strengths: Identify significant issues (e.g., long wait times, poor hygiene, safety concerns) and standout features (e.g., exceptional cuisine, distinctive ambiance, or unique menu options).

Output formatting instructions:

The rating is from 1 to 5, higher the better. 3 is average. 4 and 5 means good and excellent. 2 means below average, 1 means bad. Be faithful to the review's statement and give a rating accordingly from 1 to 5.

1. Evaluate the flavor of the dishes on a scale of 1 to 5.

2. Evaluate the freshness of the food on a scale of 1 to 5.

3. Evaluate the service of the restaurant in general with a scale of 1 to 5, considering waiting time, service, and any interaction between the guest and the staff.

4. Evaluate the environment of the restaurant from 1 to 5. Including the cleanliness of the restaurant, the kitchen, the surroundings, as well as the decorations and vibes of the restaurant. The better the environment, the better the score.

5. Evaluate the value of the restaurant from 1 to 5. If it is overly priced then it will have a lower score. If it's closer to transportation and other attractions then it might have a higher score.

—— Example 1 starts ——

This place has a rating of 2 for flavor. The food is often described as bland and mediocre, with many reviewers noting that it lacks seasoning and freshness.

This place has a rating of 2 for freshness. Several reviews mention old or wilted produce, and issues with food being served cold or not freshly prepared.

This place has a rating of 2 for service. Service is frequently criticized for being slow, inattentive, or unprofessional, with multiple reports of staff ignoring customers or being rude.

This place has a rating of 3 for environment. The diner has a clean and modern decor, but the ambiance is often described as awkward or uncomfortable due to the staff's behavior and the music choice.

This place has a rating of 2 for value. Prices are considered high for the quality of food served, leading many to feel that they are not getting good value for their money.

—— Example 1 Ends ——

—— Example 2 Starts ——

This place has a rating of 4 for flavor. The food generally receives praise for its flavor, with standout dishes like the brown butter ravioli and khachapuri being frequently mentioned. However, some dishes were noted as mediocre or lacking in flavor.

This place has a rating of 4 for freshness. Many reviews highlight the freshness of ingredients, particularly in salads and seafood dishes. The house-baked focaccia and pastries are also noted for their quality.

This place has a rating of 3 for service. Service experiences are mixed, with some diners reporting attentive and friendly staff, while others encountered slow service and disorganization. The inconsistency in service quality is a recurring theme.

This place has a rating of 5 for environment. The restaurant's decor and ambiance receive high praise, described as beautiful, modern, and inviting. The spacious layout and natural lighting contribute to a pleasant dining experience.

This place has a rating of 3 for value. While some diners feel the prices are justified by the quality of food and ambiance, others find the portions small and the overall experience not worth the cost, leading to a mixed perception of value.

—— Example 2 Ends ——

Given reviews: {reviews}

Your evaluation:

## D.2    Human Query Generation

The following prompt asks LLM to generate a human-like query based on the input preference list.

Craft a a human like query for a travel plan given the following information. The input includes details such as trip duration, budget type, attractions types that the traveler wants to visit, dining preferences that they want to try, and accommodation requirements. Make sure each pairs of key words, like good environment, good location, are mentioned specifically.

—– Example Starts —–

Input:

- general: 2 days, moderate budget,

- attraction: history oriented,

- restaurants: French, good environment,

- hotel: good quality, good location

Output: I want to go for a 2-day trip with a moderate budget. I want to visit some history-oriented attractions. Please find some good environment restaurants that provide French cuisine, I want to stay in a good quality hotel in a good location.

—– Example Ends —–

PromptInput: {input}

Output:

## D.3  No Route Optimization Prompt

Use this prompt for all tasks that don't request route optimization. In our design, only Task 1 uses this prompt.

You are a proficient travel planner. Based on the given information and query, you will generate a travel plan like the following example. Ensure that all recommendations and their addresses are organized in chronological order for each day. Give exactly 4 attraction recommendations for each day. Be considerate, concise and well-structured.

—– Example Starts —–

Query: I am planning a 2-day trip with an expensive budget. I would like to visit some history-oriented attractions. Please recommend Japanese restaurants with a good environment. For accommodation, I am looking for a hotel with good location, good quality, and good service.

Travel Plan:

Day X:

- Accommodation: - Name: XXXX Address: XXXX, XXXX

- Breakfast: - Name: XXXX Address: XXXX, XXXX

- Morning Attraction: - Name: XXXX Address: XXXX, XXXX

- Lunch: - Name: XXXX Address: XXXX, XXXX

- Afternoon Attraction: - Name: XXXX Address: XXXX, XXXX; - Name: XXXX Address: XXXX, XXXX

- Dinner: - Name: XXXX Address: XXXX, XXXX

- Night Attraction: - Name: XXXX

—– Example Ends —–

Given Information: {given_information}

Query: {query}

Travel Plan:

### D.4 Route Optimization Prompt

Both Task 2 and Task 3, which ask for route optimization, use this prompt. The difference is that Task 2's given information is all data, while Task 3's given information is filtered data based on preference and the spatial clustering algorithm. The planner module in the Tool-Use mode also uses this prompt.

> You are a proficient travel planner. Based on the given information and query, you will generate a travel plan like the following example. Ensure that all recommendations and their addresses are organized in chronological order for each day. Give exactly 4 attraction recommendations for each day. Be considerate, concise and well-structured. Please also optimize the routes for the trip. For each day, find attractions that are close to each other for the recommendations.
>
> —— Example Starts ——
>
> Query: I am planning a 2-day trip with an expensive budget. I would like to visit some history-oriented attractions. Please recommend Japanese restaurants with a good environment. For accommodation, I am looking for a hotel with good location, good quality, and good service.
>
> Travel Plan: Day X:
>
> - Accommodation: - Name: XXXX Address: XXXX, XXXX
>
> - Breakfast: - Name: XXXX Address: XXXX, XXXX
>
> - Morning Attraction: - Name: XXXX Address: XXXX, XXXX
>
> - Lunch: - Name: XXXX Address: XXXX, XXXX
>
> - Afternoon Attraction: - Name: XXXX Address: XXXX, XXXX; - Name: XXXX Address: XXXX, XXXX
>
> - Dinner: - Name: XXXX Address: XXXX, XXXX
>
> - Night Attraction: - Name: XXXX
>
> —— Example Ends ——
>
> Given Information: {given_information}
>
> Query: {query}
>
> Travel Plan:

### D.5 Tool use: ReACT Prompt

Here is the prompt inspired by ReACT (Yao et al., 2022) and TravelPlanner (Xie et al., 2024).

> Collect information for a query plan using interleaving 'Thought', 'Action', and 'Observation' steps. Ensure you gather valid information related to transportation, dining, attractions, and accommodation. All information should be written in Notebook, which will then be input into the Planner tool. Note that the nested use of tools is prohibited. Don't include phrases like "Action: ", "Action 5", "Thought 1", or "Thought: "in your response. 'Thought' can reason about the current situation, and 'Action' can have 5 different types:
>
> (1) AccommodationSearch[Budget,Preference]:
>
> Description: Find the accommodation that matches the preference.
>
> Parameters:
>
> Budget: The budget mentioned in the query.
>
> Preference: A list of preferences mentioned in the query.

Example: AccommodationSearch[Moderate Budget,[Good Location, Good Service]] would return the moderate price hotel that has a good or excellent location, as well as a good or excellent service.

(2) AttractionSearch[Budget, Preference]:

Description: Find the attractions that matches the preference.

Parameters:

Budget: The budget mentioned in the query.

Preference: A list of preferences mentioned in the query.

Example: AttractionSearch[Cheap budget,[Nature Oriented]] would return the cheap price and nature - oriented attractions.

(3) RestaurantSearch[Budget, Cuisine, Preference]:

Description: Find the restaurants that matches the preference.

Parameters:

Budget: The budget mentioned in the query.

Cuisine: The cuisine mentioned in the query.

Preference: A list of preferences mentioned in the query.

Example: RestaurantSearch[Expensive budget, Vietnamese, [Good Flavor, Good Value]] would return the expensive restaurants that offer Vietnamese cuisine, with good or excellent flavor and good or excellent value.

(4) BusinessClusterSearch[]:

Description: A tool that finds the number of business clusters given the information that you've collected. The tool will choose what business to be considered and return their spatial clustering information.

Example: BusinessClusterSearch[] would return you a list of business clusters among some attractions and hotels that you've collected. The businesses in the same cluster indicates that they are closer to each other and prefered to be arranged for the same day of the travel.

(5) Planner[Query]

Description: A smart planning tool that crafts detailed plans based on user input and the information stored in Notebook.

Parameters:

Query: The query from user.

Example: Planner[Give me a 3-day trip plan in Philadelphia] would return a detailed 3-day trip plan.

You should use as many as possible steps to collect engough information to input to the Planner tool.

Each action only calls one function once. Do not add any description in the action. Do not start action with "1. ", state the action directly.

Query: {query}{scratchpad}

### D.6 Itinerary Entry Extraction Prompt

Extract the travel itinerary and parse the businesses' information into the JSON format as below. Be faithful and concise. Correctly document the right number of the attractions. Only write down the name and address of the businesses. If certain recommendations (like meals or accommodations) are not provided, replace the information with "-" for name and address. If recommendations for a session of attraction is not provided, replace the information as an empty array.

# E    Algorithms

## E.1    Total Distance Gap Algorithm

We adapted the classic Traveling Salesmen Problem (Hoffman et al., 2013) algorithm to allow the evaluation between multiple days and different returning points. This algorithm can calculate the optimized routes when the returning hotel is different each night.

The main adaptation happens in line 18, where multiple hotel returns across the trip is allowed and the calculation of plan wise TSP is made possible. The main limitation currently is the TSP algorithm have limits about the number of node in the calculation. Gemini tends to recommend more than 5 attractions per day make the evaluation not possible with current algorithm.

```python
def totalCost_Multiday(mask, pos, day, cordinates, n, visited, cost, info_lists, memo):
    visit_requirement = len(cordinates[day])
    distance_list = []
    i_list = []

    # Get which hotel need to return to for current day
    hotel_index = getHotelIndex(day,cordinates)

    # Base case: if all cities are visited, return to hotel for current day
    if mask == (1 << n) - 1:
        return cost[pos][hotel_index]

    # Memorization
    if memo[pos][mask] != -1:
        return memo[pos][mask]

    # Main Adapatation: This condition check allows returned to different hotels and break down
        days in plan
    if visit_requirement == visited:
        for i in range(n):
            if (mask & (1 << i)) == 0:
                i_list.append(i)
                distance_list.append(
                    cost[hotel_index][i] + totalCost_multiday(
                        mask | (1 << i), i, day + 1, cordinates, n, 2, cost, info_lists, memo
                    )
                )

        info_list = [pos, i_list, distance_list]
        info_lists.append(info_list)

        return min(distance_list) + cost[pos][hotel_index] # change this to the old hotel position

    # Try visiting every city that has not been visited yet
    for i in range(n):
        if (mask & (1 << i)) == 0:
            i_list.append(i)
            # If city i is not visited, visit it and update the mask
            distance_list.append(
                cost[pos][i] + totalCost_multiday(
                    mask | (1 << i), i, day, cordinates, n, visited + 1, cost, info_lists, memo
                )
            )

    # Store an info_list to retrieve the optimized order
    info_list = [pos, i_list, distance_list]
```

```
    info_lists.append(info_list)

    memo[pos][mask] = min(distance_list)

    return min(distance_list)
```

