# OpenReview forum: "ItinBench: Benchmarking Planning Across Multiple Cognitive Dimensions with Large Language Models"
_TMLR — Under review for TMLR_

### Review · Reviewer_tmiq · 2026-06-23

**Summary Of Contributions:**

The paper introduces ItinBench, a single-city (Philadelphia, Yelp-sourced) trip-planning benchmark that augments the usual verbal-reasoning evaluation (preference satisfaction over a POI pool) with a spatial-reasoning evaluation (day-wise and plan-wise route optimization), scored against an adapted multi-day/multi-hotel TSP oracle. Four tasks vary the information given to the model, which lets the benchmark probe verbal and spatial demands in isolation and in combination. Five models are evaluated (Llama 3.1 8B, Mistral Large, Gemini 1.5 Pro, GPT-4o, o1), supported by a small human study and one qualitative case study. The most valuable idea is the reframing of "LLM spatial reasoning" as possibly semantic manipulation of textualized spatial relations rather than geometric cognition, together with the two-level decomposition in the case study (models can form correct first-level clusters but fail to organize clusters into a globally efficient route). The adapted TSP harness for variable return-hotels is a concrete, reusable technical artifact. Main weaknesses: single-run evaluation with no variance, single city, an evaluation harness that drops non-conforming models, and a mechanistic claim that outruns its evidence.

**Additional Comments:**

The writing needs a careful pass: recurring typos ("Salemens" for Salesmen, "oritentaion," "cordinates," "complied" for compiled, "engough"), and Figure 1 is dense enough to be hard to parse as the paper's overview. None of this affects the science, but it currently lowers confidence in the reported details.

**Audience:**

Yes

**Audience Explanation:**

Benchmark design for LLM planning, the verbal/spatial trade-off, and the "are we measuring spatial cognition or text manipulation?" question are squarely of interest to TMLR readers working on agents, reasoning, and evaluation. The honesty of the Limitations section (it raises the propositional-shortcut concern itself) is a plus.

**Broader Impact Concerns:**

No substantive ethical concerns. Two minor points: (a) confirm that redistributing Yelp-derived data (even ratings-only) is permitted under the dataset/Fusion API terms, and document the license in the paper; (b) note representativeness limits, a single US city, English-only, real named businesses, so readers do not over-generalize the planning conclusions.

**Claims And Evidence:**

No

**Claims Explanation:**

TMLR's bar is whether claims match the evidence, and here there is an addressable gap.

Two issues:
- Section 3.4 states results come from "a single run" at temperature 1, with no seeds, repetitions, variance, or significance tests. The headline numbers the conclusions rest on, e.g., Total-DG falling from ~25% to ~15% when clusters are provided, or o1's verbal lead shrinking under spatial load, cannot be distinguished from run-to-run noise as presented. For a benchmark paper whose claims are comparative, this undercuts "accurate, convincing" evidence.
- Claim-evidence mismatch on mechanism. The behavioral claim ("models only improve route quality when cluster information is supplied as text") is reasonably supported. But the stronger mechanistic claim ("this shows reliance on semantic shortcuts rather than genuine spatial cognition") is not isolated: an equally consistent reading is a capability gap (models cannot parse raw lat/long geometry from text at all), which is about what fails, not which mechanism is used. The supporting case study is $n=1$.

Both are fixable: add repeated runs with reported variance/CIs and re-check which gaps survive; and either scope the mechanistic language down to an observation, or add a direct experiment (below) that turns the n=1 intuition into a measured result. With either fix I would move this to Yes.

**Requested Changes:**

Some major points that should be fixed:
- Run each (model, task) multiple times and report mean with variance (or bootstrap CIs); state explicitly which differences are robust to noise. Single-run numbers are the main blocker.
- Convert the central spatial hypothesis into a measurable test: add a "clusters given — order the clusters" condition that isolates higher-level route organization from first-level grouping. This directly probes the case-study claim across the dataset rather than via one example.
- Fix the evaluation harness so non-conforming outputs are scored rather than dropped. Gemini's spatial rows are "-" purely because it proposes $>5$ attractions and the exponential bitmask-DP TSP (Appendix E.1) cannot handle it. Consider an approximate solver (e.g., OR-Tools / LKH) to remove the node cap and the missing-data bias.
- Resolve the decoding contradiction: "All temperatures are set to 1, except for OpenAI o1, which is set to 1" (Sec. 3.4) is self-contradictory; specify exact settings and whether o1 reasoning effort was controlled.

Minor points:
- Add at least one additional city or an OOD split, the "real-world planning" framing invites multi-city, and current claims generalize from one city.
- Refresh/expand the model set; the lineup is dated for a current submission, and adding a recent reasoning model would test whether the trade-off conclusions hold.
- Clarify ECJ: a baseline of 86.2 for the greedy heuristic and values >100% for LLMs are unintuitive; define bounds and units.
- Justify the Macro 75% threshold given that preference counts vary (6–10 per query; 1–3 for restaurants/hotels).
- Sharpen the trade-off result: the verbal-vs-spatial interaction is the paper's most interesting effect but is shown mostly anecdotally (o1's shrinking lead); report it as a controlled contrast.

---

### Review · Reviewer_cwmk · 2026-07-01

**Summary Of Contributions:**

**Summary**
The paper proposes ItinBench, a new spatial-reasoning benchmark for LLMs. The authors argue that the majority of the benchmarks focus on verbal reasoning, so they expand by creating a benchmark on spatial reasoning using Yelp locations and user reviews for a city. They attempt to capture user preferences by creating queries based on the reviews and also perform optimal itinerary planning (hence “ItinBench”). They have variants of experiments: with or without optimization, with or without domain knowledge (i.e. preference-filtered data, clusters of locations, tool use).

They evaluate 5 LLMs (variants of LLaMa-3.1, GPT-4o, Gemini-1.5 Pro, Mistral, OpenAI o1) on various metrics: out-of-pool, missing information, validation rate, etc. for verbal; distance-gap, cluster-jump, etc. for spatial reasoning. Overall findings are that LLMs are not able to perform well without filtered data, and without clustering information, and even with those they are far from perfect. They also perform human evaluations to show that the preferences match the evaluation metrics (validation rate) and that filtered data and tool-use based results are more aligned with human preferences.

**Strengths**
1. The authors attempt to expand the evaluation of LLMs towards spatial reasoning, which is a right step to expand the evaluation to more domains.
2. The experiments are sound and the metrics are well-designed to evaluate specific aspects of the LLMs (verbal vs spatial reasoning ability).
3. The writing is clear to understand and tables are generally understandable.

**Weaknesses**
1. The authors claim that their benchmark evaluates under “real-world” conditions and combines multiple reasoning domains, but they only evaluate LLMs on one additional dimension. Moreover, the task that they create is not realistic, they stick to specific types of places, focus only on one city, focus only on data from one application, and limit the attributes significantly. Realistic planning is actually way more complicated, where opening times, closing times, spot availability, dietary preferences, permits, “crowded”ness, best time to go (e.g. sunset), etc. play a huge role in itinerary planning. The benchmark captures only simplistic cases. The fact that a greedy algorithm is able to achieve perfect performance is also a sign that this benchmark is not realistic.
The authors do not discuss how their benchmark is different from any other spatial reasoning benchmark in detail. They discuss TripTailor briefly, but from the reading it looks like the only new thing ItinBench adds is “algorithmic evaluation”, which is just TSP and some other baselines.
2. The TSP evaluation is also flawed, how does TSP consider the human preferences when computing the optimal routes? It is hard to know whether LLMs are incorrect by just looking at aggregate numbers. Perhaps LLM attempts to incorporate human preferences when planning.
3. Details on evaluation are lacking - how do they measure which information is missing or out-of-pool - is a human involved? Is this judged by an LLM?
4. Figure 2 and 4 are too small to read easily. The font is extremely tiny.
5. Design of experiments: “set to 1, except for OpenAI, which is set to 1.” - looks like this was something else but is it incorrect?

**Questions**
1. What are the features used for clustering algorithms?
2. Did the authors ever expand to MLLMs and try to use map locations, or agentic LLMs where they have access to browser / maps, etc? State of the art agentic LLMs are able to create reasonable itineraries and I believe this benchmark might be too simple for them.

**Audience:**

No

**Audience Explanation:**

The primarily findings of the paper can be summarized as:
- LLMs are not great at spatial reasoning.
- LLMs improve when the spatial reasoning task is accompanied by domain knowledge (filtered data, clusters, etc.) or the scope of the problem is reduced.
- GPT-4o / OpenAI o1 is the best performing model.

To me, it feels like a lot of the findings are intuitive. The idea is overall nice, and the experiments are sound, but findings seem limited to a watered-down version of the problem.

**Broader Impact Concerns:**

No ethical concerns as such.

**Claims And Evidence:**

No

**Claims Explanation:**

I would say that the main claims as follows:
- LLMs are not great at spatial reasoning: This claim is supported by the authors experiments.
- Converting to verbal reasoning (using tool-use, clusters or filtering the data) helps: This claim is also supported by the authors experiments.
- ItinBench does a better job of evaluating spatial reasoning / is more real-world than other benchmarks: Whether this claim is supported is arguable, I feel the benchmark is too simplistic and while LLMs still don't do a great job on it, the task and the evaluation feel simple to me, very limited in scope, and a simple extension of previous works.
- The authors expand the domains on which LLM reasoning is evaluated: Spatial reasoning has already been evaluated previously, and the authors just evaluate on the same domain in a different way with a different evaluation algorithm.

**Requested Changes:**

1. The metrics description takes up a lot of the space on the paper. Instead, this space could be utilized for adding more interesting findings, breakdowns of the failure modes, interesting things like CoT  - maybe ask the LLM why it made decision A when optimized algorithm suggests decision B? I think some more analysis would definitely help me push this paper for acceptance.
2. How do MLLMs perform? Can the map be provided with the locations marked, will it do a good job? What about agentic LLMs? Not necessary, but strengthens the paper and shows due diligence.
3. Is is possible to include more cities, more kinds of destinations to make the benchmark more general? Can we add more attributes to make the benchmark more realistic? Any attempt here would strengthen the paper.

---

### Review · Reviewer_jQGC · 2026-07-16

**Summary Of Contributions:**

This paper introduces ItinBench to examine whether LLMs can jointly handle verbal reasoning and spatial reasoning in trip itinerary planning. It evaluates preference and constraint satisfaction alongside route optimisation using four task settings and an adapted TSP solver. The results show that current LLMs struggle to maintain consistent performance across both dimensions, while filtered data and explicit spatial cluster information improve planning performance.

***Strengths:***

- The benchmark jointly evaluates preference-constrained recommendation and route optimisation, which better reflects the multiple requirements involved in practical itinerary planning.
- The use of algorithmic route metrics and human evaluation provides complementary evidence beyond judging only the fluency or apparent plausibility of generated plans.

***Weaknesses:***

- The current findings remain relatively coarse-grained, although the benchmark aims to diagnose performance across different cognitive dimensions. Candidate filtering and cluster information are introduced together, so their individual contributions cannot be determined.
- The task does not fully isolate spatial reasoning from retrieval, coordinate interpretation, constraint satisfaction, and combinatorial optimisation. The results therefore demonstrate difficulty with spatially constrained planning, but do not directly establish a general limitation in spatial cognition.
- The benchmark is constructed only for Philadelphia. Since urban layout, POI density, and travel patterns may differ substantially across cities, the broader conclusions about LLM planning and spatial reasoning appear somewhat premature.
- The claimed trade-off between verbal and spatial reasoning is based on a single generation per query at temperature 1, without repeated runs or uncertainty estimates.
- The human evaluation supports the overall validity of the benchmark, but the analysis relies mainly on similar ranking trends and does not report inter-annotator agreement or formal statistical alignment.

**Additional Comments:**

N/A

**Audience:**

Yes

**Audience Explanation:**

The paper may interest researchers working on LLM planning, tool-using agents, and benchmark design.

**Broader Impact Concerns:**

None. The paper adequately addresses data licensing and the handling of user-generated reviews.

**Claims And Evidence:**

No

**Claims Explanation:**

The experiments may support the broad finding that current LLMs struggle to jointly satisfy detailed preferences and produce efficient routes. However, the stronger claims concerning spatial cognition, semantic shortcuts, and trade-offs between cognitive dimensions rely on confounded task settings and limited statistical evidence. The claims should therefore be narrowed.

**Requested Changes:**

- Separate candidate filtering and cluster information through controlled ablations.
- Repeat the main experiments and report uncertainty or statistical significance.
- Narrow the claims about spatial cognition, semantic shortcuts, and reasoning trade-offs.
- Report inter-annotator agreement and quantitative alignment with the automatic metrics.
- Add a baseline combining LLM-based selection with an external route solver.
- Discuss generalisation beyond a single city more explicitly.